# Time-Restricted Eating Without Exercise Enhances Anaerobic Power and Reduces Body Weight: A Randomized Crossover Trial in Untrained Adults

**DOI:** 10.3390/nu17183011

**Published:** 2025-09-20

**Authors:** Zifu Yu, Takeshi Ueda

**Affiliations:** Graduate School of Humanities and Social Sciences, Hiroshima University, Higashihiroshima 739-8524, Japan; d211386@hiroshima-u.ac.jp

**Keywords:** time-restricted eating (TRE), aerobic endurance, anaerobic power, body weight, non-exercise intervention

## Abstract

**Background**: Time-restricted eating (TRE), a dietary strategy that confines daily food intake to specific time windows, has been shown in animal models to enhance physical performance even without exercise training. However, evidence in humans under non-exercise conditions remains limited. **Objective**: This study aimed to investigate the effects of early TRE (eTRE; 08:00–14:00) and delayed TRE (dTRE; 12:00–18:00) on body weight, aerobic endurance, and anaerobic power in untrained adults. **Methods**: In a randomized crossover trial, 28 healthy university students (mean age 23.47 ± 2.87 years; 16 women) completed two 4-week interventions, eTRE and dTRE, separated by a 2-week washout. Participants did not engage in any structured exercise during the intervention period. Body weight, aerobic endurance (cycling time to exhaustion at a fixed workload), and anaerobic power (peak power output during sprint cycling) were assessed after each phase. **Results**: Body weight significantly decreased after eTRE (−1.56 kg; 95% CI [−2.07, −1.05]; *p* < 0.001; Cohen’s $d_{z}$ = 1.17) and dTRE (−0.61 kg; 95% CI [−1.12, −0.10]; *p* = 0.022; Cohen’s $d_{z}$ = 0.55), with a greater reduction observed in eTRE compared to dTRE (−0.95 kg; 95% CI [−1.74, −0.16]; *p* = 0.020). Aerobic endurance showed no significant change across phases (all *p* > 0.05). Anaerobic power significantly improved after both eTRE (+21.25 W; 95% CI [12.03, 30.47]; *p* < 0.001; Cohen’s $d_{z}$ = 1.10) and dTRE (+35.43 W; 95% CI [26.21, 44.65]; *p* < 0.001; Cohen’s $d_{z}$ = 1.20), and the improvement was significantly greater in dTRE compared to eTRE (+14.18 W; 95% CI [1.79, 26.57]; *p* = 0.025; Cohen’s $d_{z}$ = 0.54). **Conclusions**: Both early and delayed TRE independently led to weight loss and enhanced anaerobic power in the absence of an exercise intervention. eTRE was more effective for weight reduction, whereas dTRE produced greater improvements in anaerobic performance.

## 1. Introduction

Aerobic endurance and anaerobic power are important indicators for evaluating an individual’s physical performance and metabolic function [1,2]. These capacities not only determine athletic performance but are also closely associated with general health, quality of life, and the prevention of metabolic diseases such as type 2 diabetes and cardiovascular disease [3,4]. Previous studies have shown that improvements in aerobic capacity are significantly linked to better glucose homeostasis, insulin sensitivity, lipid profiles, and cardiopulmonary function [5,6,7]. Additionally, anaerobic power has been found to be a better predictor of physical function and independence in older adults than muscle strength, underscoring its importance in healthy aging [8].

Traditionally, enhancing these physical capacities has relied on aerobic training, resistance training, and high-intensity interval training (HIIT). However, in recent years, nutritional strategies—particularly meal timing—have garnered increasing attention for their role in optimizing physical performance and metabolic health. Time-restricted eating (TRE), an emerging dietary intervention that limits daily food intake to a 4–12 h window, often leads to a spontaneous reduction in caloric intake by approximately 20% [9], even though no explicit calorie restriction is imposed. By extending fasting duration, TRE activates various metabolic adaptation mechanisms, including enhanced insulin sensitivity and glucose tolerance [10,11,12], improved mitochondrial function [13,14], increased lipolysis and fat oxidation [15,16], and alignment of circadian clock gene expression across peripheral tissues such as the liver and skeletal muscle [10,17,18]. These adaptations contribute to improved energy metabolism, reduced cardiometabolic risk, and potentially enhanced physical performance. [9,10,11,12,18,19,20,21,22,23,24,25].

Current animal and human studies suggest that TRE may enhance physical function through three primary mechanisms. First, TRE induces a “metabolic switch”, shifting energy substrates from glucose to fatty acids and ketones, which has been shown—primarily in animal models—to promote mitochondrial biogenesis, fat oxidation, and cellular repair processes [24,26].

Second, TRE strengthens circadian rhythm synchronization by consolidating food intake within a consistent daytime window. Circadian rhythms, which are regulated by central and peripheral clocks, play a vital role in coordinating metabolic processes such as insulin sensitivity, glucose uptake, and hormonal secretion [10,27,28,29]. Both animal [30] and human studies [13] suggest that aligning food intake with circadian rhythms—particularly by avoiding late-night eating—can improve energy metabolism, reduce inflammation, and enhance skeletal muscle efficiency.

Third, TRE may activate autophagy, a conserved intracellular recycling process responsible for clearing damaged organelles and proteins [11,31]. This mechanism has been robustly demonstrated in animal studies [14,32], while human evidence remains indirect. Fasting-induced autophagy in metabolically active tissues like skeletal muscle may facilitate mitochondrial quality control and oxidative stress resistance, potentially contributing to improved physical performance.

Existing human studies on TRE have predominantly focused on its metabolic benefits in overweight and obese populations, demonstrating consistent improvements in body weight, blood glucose, lipid profiles, and insulin sensitivity [9,12]. In parallel, a separate body of research has investigated whether combining TRE with structured exercise training can further enhance muscle adaptation and physical performance. While some studies have reported modest gains in strength and endurance, the overall evidence suggests that the performance-enhancing effects of TRE in trained individuals are limited, with benefits more prominent in body composition and metabolic flexibility [15,33,34].

Many studies exploring the effects of TRE on physical performance have used Ramadan fasting as a natural model [35,36,37]. However, findings remain inconsistent. Notably, this form of fasting is misaligned with the circadian rhythm, as it prohibits food and water intake during the day, which may adversely affect exercise performance [38].

While these studies provide useful insights, they often include structured exercise interventions. It remains unclear whether TRE alone—without exercise—can enhance physical performance through modulation of circadian rhythms, autophagy, and metabolic pathways. Theoretically, fasting-induced metabolic switching, rhythm alignment, and cellular repair mechanisms may confer physiological benefits even in the absence of exercise stimuli, particularly among untrained individuals who may exhibit greater adaptive potential due to their lower baseline fitness levels.

Therefore, this study aims to fill this knowledge gap by systematically evaluating whether TRE can improve aerobic endurance and anaerobic power in healthy adults under non-exercise conditions. Recent studies have suggested that the timing of the eating window during the day may differentially affect metabolic outcomes [29,39,40,41,42,43,44]. In light of this, we incorporated two distinct TRE protocols into our study design: early TRE (eTRE), with food intake restricted to the earlier part of the day, and delayed TRE (dTRE), with food intake shifted to later hours.

This approach not only allows us to examine the overall efficacy of TRE on physical performance but also enables a direct comparison between different timing strategies.

Based on prior rodent studies demonstrating significant improvements in physical performance following TRE [45], we hypothesize that TRE may similarly enhance aerobic endurance and anaerobic power in untrained individuals, even in the absence of a structured exercise intervention.

## 2. Methods

This randomized crossover trial was designed and reported in accordance with the CONSORT 2010 guidelines, including its extension for crossover trials. A completed CONSORT checklist is provided in the Appendix A.

### 2.1. Participants

Participants were recruited through university-affiliated social media platforms, targeting healthy, non-obese (BMI < 30) individuals. The inclusion criteria were as follows: (1) no history of cardiovascular, metabolic, or musculoskeletal disorders; (2) no current smoking or regular alcohol consumption; (3) no participation in any structured exercise program during the study period; and (4) willingness and ability to adhere to all study protocols.

A total of 39 individuals were initially screened. After a detailed interview process, 28 university students (12 males and 16 females) voluntarily participated in the study. Prior to participation, all subjects provided written informed consent.

This study was approved by the Ethics Committee of the Graduate School of Humanities and Social Sciences, Hiroshima University (Approval No. HR-ES-001701, approved on 21 May 2024) and was conducted in accordance with the principles outlined in the Declaration of Helsinki.

### 2.2. Study Design

A 10-week randomized crossover design was implemented to examine the effects of eTRE and dTRE on body weight, aerobic endurance, and anaerobic power. As illustrated in Figure 1, all participants first underwent baseline assessments (T1) before being randomly assigned to either the eTRE or dTRE condition for a 4-week intervention period.

Randomization was performed using a computer-generated random sequence (simple randomization with a 1:1 allocation ratio). The sequence was generated by an independent researcher not involved in data collection or analysis. Participants were sequentially assigned to groups based on this predetermined sequence.

Following the second round of testing (T2), a 2-week washout period was introduced, after which participants crossed over to the alternate TRE condition for another 4 weeks. The final assessments were conducted at the end of the study period (T3).

### 2.3. Dietary Intervention and Compliance

Participants followed two distinct TRE protocols, each lasting 4 weeks, separated by a 2-week washout period. In both conditions, a 6 h daily eating window was prescribed, and only water was permitted outside the designated window.

eTRE Phase:First meal between 08:00 and 09:00; last meal between 14:00 and 15:00.dTRE Phase: First meal between 12:00 and 13:00; last meal between 18:00 and 19:00.

Total energy intake (kcal) was recorded over three consecutive days at three time points: baseline, during the eTRE phase, and during the dTRE phase. Participants were instructed to photograph all meals and beverages using their smartphones and submit the images to the research team. Caloric intake was estimated by trained researchers using a nutrition-tracking application (Asken, Asken Inc., Shinjuku-ku, Japan).

To ensure protocol adherence, participants were instructed to maintain consistent lifestyle habits, including sleep schedules, physical activity levels, and dietary composition. They were asked to refrain from strenuous exercise, alcohol consumption, or any major lifestyle changes during the intervention period. Weekly online check-ins were conducted to monitor adherence and provide support. In addition, participants completed weekly self-report logs that documented their sleep duration, daily activity (e.g., commuting, walking), and any notable deviations in diet. These measures helped ensure that observed outcomes were attributable to the intervention rather than external behavioral fluctuations.

### 2.4. Outcome Measures

#### 2.4.1. Body Weight

Body weight was measured in a fasted state between 06:30 and 08:30 on each test day (T1, T2, T3) using a calibrated electronic scale (Tanita MC-780A, Tanita Corporation, Tokyo, Japan) with a precision of ±0.1 kg. Participants were instructed to refrain from eating after 22:00 the previous night (water permitted), to empty their bladder before measurement, and to wear light clothing.

#### 2.4.2. Anaerobic Power

Anaerobic performance was evaluated using a mechanically braked cycle ergometer (Monark Ergomedic 894E, Monark Exercise AB, Vansbro, Sweden). Each participant performed three 10 s all-out sprints, with 2 min rest intervals between efforts. The highest peak power output (in watts) recorded across the three sprints was used for analysis. This short-duration, all-out sprint protocol using peak power output is a widely accepted standard method in exercise physiology for assessing anaerobic capacity [46]. Following this test, participants rested for 30 min before undergoing the aerobic assessment. Only water intake was permitted during this period.

#### 2.4.3. Aerobic Endurance

Aerobic endurance was assessed using a constant-load cycling test on the same ergometer, a widely used protocol in exercise physiology for evaluating endurance performance and intervention effects [47]. The workload was standardized at 150 W for males and 90 W for females. Participants were instructed to maintain a cadence of 60 revolutions per minute (rpm) for as long as possible. The test was terminated when a participant failed to sustain the target cadence for more than 10 consecutive seconds. Total cycling duration (in minutes) was recorded as the outcome measure.

All physical performance assessments were conducted between 15:00 and 17:00 to control for potential circadian influences. Prior to the intervention, a familiarization session was held to ensure consistent performance and proper technique. During this session, seat height and positioning were individually adjusted and recorded for each participant, and the same configuration was applied during all test sessions to minimize intra-individual variability. All assessments were administered by the same trained research personnel, who provided standardized verbal encouragement to promote maximal effort and maintain consistency across trials.

### 2.5. Statistical Analysis

Data normality was assessed using the Shapiro–Wilk test. Endurance and power variables were normally distributed across all phases, whereas body weight deviated from normality in some phases. Given the robustness of linear mixed-effects models (LMMs) to violations of normality assumptions, LMMs were employed to analyze all outcomes.

To evaluate changes in body weight, aerobic endurance, and anaerobic power across the three experimental phases (Pre, eTRE, and dTRE), each LMM included phase as a fixed effect and participant ID as a random intercept to account for within-subject variability inherent in the crossover design. When a significant main effect of phase was detected (p≤0.05), pairwise post hoc comparisons were performed using paired-sample *t*-tests with Bonferroni correction (Pre vs. eTRE, eTRE vs. dTRE, Pre vs. dTRE). Effect sizes were estimated using Cohen’s $d_{z}$ for paired samples and interpreted as small (d=0.2), medium (d=0.5), or large (d=0.8).

All statistical analyses were conducted in IBM SPSS Statistics, version 27.0 (IBM Corp., Armonk, NY, USA). Data are presented as means ± standard deviations. All tests were two-tailed with a significance threshold of p≤0.05.

The sample size was determined based on the number of participants successfully recruited. A post hoc power analysis revealed a statistical power of 78.7% (close to 80%), suggesting that the sample size was generally sufficient to detect moderate effects.

## 3. Results

### 3.1. Participant Characteristics and Summary of Outcomes

A total of 28 participants completed the entire experimental protocol. Their baseline characteristics, including age, height, body weight, and BMI, are summarized in Table 1. In addition, Table 2 provides an overview of the observed changes in body weight, aerobic endurance, anaerobic power, and energy intake across the three experimental phases.

### 3.2. Body Weight

A linear mixed-effects model revealed a significant main effect of phase on body weight (p<0.001). Pairwise comparisons showed a significant decrease in body weight from Pre to post-eTRE (mean difference = −1.56 kg; 95% CI [−2.07, −1.05]; p<0.001; Cohen’s $d_{z}$ = 1.17) and from Pre to post-dTRE (mean difference = −0.61 kg; 95% CI [−1.12, −0.10]; p=0.022; Cohen’s $d_{z}$ = 0.55). A greater reduction was observed during the eTRE phase compared to the dTRE phase (mean difference = −0.95 kg; 95% CI [−1.74, −0.16]; p=0.020; Cohen’s $d_{z}$ = 0.56). These results indicate that body weight decreased in both the eTRE and dTRE phases compared to baseline, with a more pronounced effect in the eTRE phase (Figure 2).

### 3.3. Aerobic Endurance

There were no significant differences in aerobic endurance across the three phases (p>0.05). Pairwise post hoc comparisons likewise revealed no statistically significant differences between any of the phases (all p>0.05), and the effect sizes were negligible (Cohen’s $d_{z}$ < 0.2). Aerobic endurance remained stable throughout the study period across all phases (Figure 3).

### 3.4. Anaerobic Power

The linear mixed-effects model revealed a significant main effect of phase on anaerobic power (p<0.001). Anaerobic power significantly increased from Pre to post-eTRE, with a mean difference of +21.25 W (95% CI [12.03, 30.47]; p<0.001; Cohen’s $d_{z}$ = 1.10), and from Pre to post-dTRE, with a mean difference of +35.43 W (95% CI [26.21, 44.65]; p<0.001; Cohen’s $d_{z}$ = 1.20). Power output was also higher in the dTRE phase than in the eTRE phase, with a mean difference of +14.18 W (p=0.025; Cohen’s $d_{z}$ = 0.54) (Figure 4).

## 4. Discussion

TRE, a dietary approach that limits food intake to specific windows each day, has been associated with various metabolic benefits, including weight loss, enhanced insulin sensitivity, improved circadian rhythm alignment, and more efficient energy metabolism. Although animal studies have demonstrated that meal timing alone can improve skeletal muscle metabolism and endurance—even in the absence of exercise—evidence in humans under non-exercise conditions remains limited.

To address this gap, the present randomized crossover trial compared two distinct TRE protocols—early TRE (eTRE; 08:00–14:00) and delayed TRE (dTRE; 12:00–18:00)—in healthy young adults without structured physical training. We examined their effects on body weight, aerobic endurance, and anaerobic power. The findings demonstrated that both protocols resulted in weight loss, preserved aerobic endurance, and enhanced anaerobic power. Notably, eTRE led to greater reductions in body weight, while dTRE produced larger gains in anaerobic performance. To our knowledge, this is the first human crossover study to directly compare eTRE and dTRE in a non-exercising population, providing novel insights into the performance-related implications of meal timing. The following sections discuss each outcome in detail, along with possible underlying mechanisms.

### 4.1. eTRE Leads to Greater Weight Loss than dTRE

Both eTRE and dTRE significantly reduced body weight under non-exercise conditions, with eTRE resulting in greater weight loss. This supports previous evidence suggesting that earlier eating windows confer superior benefits for weight management [48]. A growing body of research has demonstrated that eTRE leads to more favorable changes in body composition and glucose metabolism compared to later TRE schedules [29]. For instance, a 5-week randomized trial in non-obese adults found that eTRE—but not mid-day TRE—significantly reduced body mass and adiposity and improved glucose control and gut microbiota [44].

Several mechanisms may account for the superior outcomes of eTRE. First, dinner tends to be more calorically dense than breakfast in typical dietary patterns. Therefore, by excluding dinner, eTRE may naturally lead to a greater reduction in total daily energy intake. Second, food intake earlier in the day aligns better with circadian rhythms governing digestive function and insulin sensitivity [11,12,49,50]. In contrast, late-day eating may promote energy storage rather than expenditure [51]. Together, these factors likely contribute to the more effective weight reduction seen with eTRE. However, since body composition was not directly assessed in this study, it remains unclear whether the observed weight loss reflected reductions in fat mass, lean mass, or water weight.

### 4.2. Aerobic Endurance Is Maintained During TRE

In our study, aerobic endurance was maintained during both the eTRE and dTRE phases, with no statistically significant improvements observed. While this might seem neutral at first glance, the ability to sustain endurance performance in the context of restricted eating windows and modest caloric reductions is physiologically relevant, especially in untrained individuals.

This finding aligns with Moro et al. (2020) [15], who conducted a 4-week TRE intervention (12:00–20:00) in elite male cyclists. Despite a rigorous training regimen and dietary control, no significant changes were observed in endurance indicators such as maximal oxygen uptake (${\mathrm{VO}}_{2\max}$) or time to exhaustion [15]. In contrast, Witt et al. (2023) demonstrated that a 4-week self-selected 16:8 TRE protocol in middle-aged male cyclists led to significant reductions in body weight (−2.5 kg), blood pressure, and fat mass, along with an increase in fat oxidation and a ∼2 min improvement in a 10 km cycling time trial [33]. Notably, these performance gains were achieved without any measurable decrease in fat-free mass or energy intake. This suggests that, under real-world conditions, TRE may enhance endurance performance in recreationally trained individuals, potentially through improved substrate utilization.

Supporting evidence from animal studies further strengthens this perspective. In a study by Chaix et al. (2014), mice subjected to time-restricted feeding without any exercise intervention exhibited a remarkable 1.8-fold increase in treadmill time to exhaustion compared with ad libitum-fed controls [45]. These improvements were not due to enhanced muscle strength or spontaneous activity levels but were instead associated with restoration of circadian rhythms and reprogramming of energy metabolism-related genes in skeletal muscle and liver. These molecular adaptations promoted more efficient switching between carbohydrate and fat metabolism, enabling a more stable energy supply during prolonged exercise.

Collectively, these findings suggest that while TRE alone may not significantly enhance aerobic endurance in highly trained individuals, it can maintain or even improve performance under certain conditions, particularly by optimizing metabolic flexibility, enhancing fat oxidation, and aligning energy metabolism with circadian rhythms. Our results further indicate that these endurance-sustaining effects may be generalizable across different eating windows (early or delayed) in untrained populations. Future studies should explore the synergistic effects of TRE combined with structured endurance training to better understand its full potential.

### 4.3. Anaerobic Power Gains Without Structured Training

In this study, participants demonstrated significant improvements in anaerobic power during both TRE phases, with a more pronounced enhancement observed during the dTRE phase. This finding is particularly noteworthy, as no systematic resistance or sprint training was conducted during the entire intervention period.

These results contrast with those of Moro et al. (2020) [15], who reported that a four-week TRE intervention did not significantly improve absolute peak power during anaerobic performance tests in young elite athletes. However, they did observe an increase in peak power relative to body weight, primarily attributed to weight loss—an outcome that is nonetheless meaningful [15]. The discrepancy between these findings may be partly explained by differences in training status. Elite athletes typically operate near their physiological limits, leaving little room for further performance enhancement. Moreover, their high weekly training volume may mask any positive effects of short-term TRE. In contrast, the untrained individuals in the present study, who lacked regular exercise habits, may have been more responsive to short-term dietary interventions and thus more likely to show measurable improvements.

A randomized crossover study by Jones et al.(2020) [21] further supports the potential of TRE to enhance muscle metabolism. They found that just two weeks of early TRE (08:00–16:00) improved skeletal muscle sensitivity to insulin and amino acids, thereby increasing the efficiency of postprandial glucose and amino acid uptake. In addition, an animal study demonstrated that TRE significantly upregulated the rhythmic expression of core circadian genes in skeletal muscle (e.g., *Bmal1*, *Per1*, *Per2*, *Cry1*), along with genes related to mitochondrial biogenesis, such as PGC-1α, *NRF1*, and *TFAM*. Electron microscopy revealed increased mitochondrial density, more compact morphology, and an enlarged cristae surface area, suggesting enhanced ATP-generating capacity [13]. These metabolic adaptations may support energy replenishment and recovery during repeated bouts of short-duration explosive activity.

The greater improvement in anaerobic performance observed during the dTRE phase may also be attributed to its feeding window (12:00–18:00), which aligns more closely with the circadian peak in neuromuscular performance. Numerous studies have reported that high-intensity exercise capacity tends to be optimal in the afternoon to early evening [52,53,54], likely due to circadian variations in sympathetic activity, core body temperature, and metabolic flexibility. Aligning nutrient intake with this peak period may result in greater pre-exercise energy availability compared to eTRE, thus facilitating enhanced performance.

In conclusion, TRE—especially dTRE—may enhance anaerobic power through multiple mechanisms, including regulation of circadian genes, promotion of mitochondrial biogenesis, increased nutrient sensitivity, and alignment of energy availability with circadian rhythms. Notably, these performance improvements were observed even in the absence of resistance training. These findings may have clinically meaningful implications. An increase of 21–35 W in anaerobic power—achieved within just 4 weeks and without exercise—represents a substantial functional enhancement, particularly for untrained individuals or populations unable to engage in structured physical activity. Given the large effect size (Cohen’s d > 0.8), such improvements may translate into better performance in daily tasks, reduced fatigue during high-effort activities, and potential benefits for rehabilitation or aging populations. Future studies should explore the sustainability of these gains, their translation into real-world functional outcomes, and the underlying mechanisms in diverse populations. Nonetheless, further studies are needed to confirm the long-term effects and generalizability of these improvements across different populations and settings.

### 4.4. Sex-Specific Considerations

Although the present study included both male and female participants (16 women and 12 men), it was not specifically designed to examine sex-related differences. Due to the limited sample size, sex-stratified analyses were not performed, and the study lacked sufficient power to detect sex-specific effects. Nonetheless, prior research suggests that men and women may respond differently to TRE, particularly regarding energy metabolism, hormonal regulation, and circadian alignment. For instance, fluctuations in estrogen and progesterone across the menstrual cycle may affect appetite and metabolic rate in women [55], potentially influencing TRE outcomes. Moreover, a recent review reported that intermittent fasting may reduce androgen markers (e.g., testosterone and free androgen index) while increasing sex hormone-binding globulin in premenopausal women with obesity—especially when food intake is limited to earlier in the day. In contrast, lean, active men may also experience reduced testosterone levels, though without adverse effects on muscle mass or strength [56]. These findings underscore the need for future studies with larger, sex-stratified samples to clarify potential sex-based differences in physiological responses to TRE.

### 4.5. Practical Implications

Our findings suggest that TRE, even in the absence of exercise, may serve as a simple and accessible strategy to enhance anaerobic power and reduce body weight. This has potential implications for individuals who are sedentary, elderly, or unable to engage in structured physical activity due to medical or lifestyle constraints. Incorporating TRE into daily routines may improve functional capacity and support metabolic health. These benefits can be especially valuable in preventive healthcare, rehabilitation programs, or as a complement to physical training interventions.

### 4.6. Limitations and Directions for Future Research

This study has several limitations. First, the participants were young, healthy adults without regular exercise habits, which may limit generalizability to other populations. Second, although a washout period was implemented, residual effects between phases cannot be ruled out, and no new baseline was established. Such residual effects could attenuate or amplify the observed effects of the second intervention, particularly if physiological or behavioral adaptations (e.g., changes in appetite regulation, circadian alignment, or insulin sensitivity) persist beyond the washout period. For example, if eTRE was administered first, improvements in metabolic parameters might carry over into the subsequent dTRE phase, potentially underestimating the relative effects of dTRE. Without re-establishing a baseline before each phase, it is difficult to fully isolate the independent effects of each intervention.

Third, the intervention duration was relatively short, and long-term sustainability remains unknown. Fourth, although the sample included both males and females, the limited sample size precluded sex-stratified statistical analyses, which may have masked potential sex-specific responses to the interventions. Fifth, detailed assessments of body composition (e.g., body mass index, fat percentage, and lean mass) were not conducted, which limits the ability to interpret potential inter-individual variability in response to TRE. Sixth, while no external control group was included, the baseline (Pre) phase—during which participants followed a normal three-meal schedule—served as an internal reference. This allowed us to evaluate changes induced by the eTRE and dTRE interventions. However, without an external control group, potential time-related confounders cannot be entirely excluded, and causal interpretations should be made with caution. Although a familiarization session was conducted prior to testing to minimize practice effects, residual learning effects cannot be entirely ruled out. Finally, dietary intake and physical activity were not tightly controlled, which may have influenced the outcomes.

Future studies should consider broader populations, extended intervention periods, larger and balanced sample sizes for subgroup analyses, and stricter control of behavioral and physiological variables—including body composition metrics—to better understand the effects of TRE.

## 5. Conclusions

This study provides evidence that both early and delayed TRE protocols elicit beneficial physiological adaptations in untrained individuals, including reductions in body weight, preservation of aerobic endurance, and improvements in anaerobic power. Notably, these effects were observed in the absence of structured exercise, underscoring the potential of meal timing as an independent modulator of physical performance. The distinct outcomes between eTRE and dTRE further suggest that aligning food intake with circadian rhythms may differentially influence endurance- versus power-oriented capacities. Future research should explore the long-term sustainability of these adaptations and investigate their interaction with formal training regimens across diverse populations.

## Figures and Tables

**Figure 1 nutrients-17-03011-f001:**
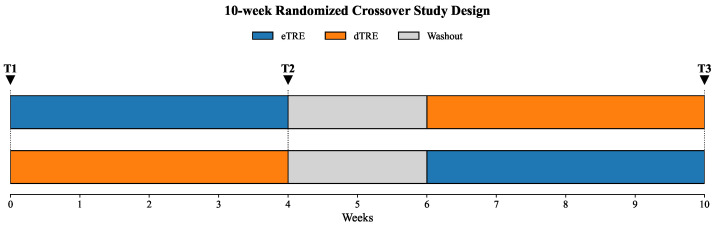
Schematic diagram of the 10-week randomized crossover trial. Participants underwent baseline testing (T1), followed by either eTRE or dTRE for 4 weeks. After T2 testing and a 2-week washout period, they crossed over to the other condition, concluding with T3 testing.

**Figure 2 nutrients-17-03011-f002:**
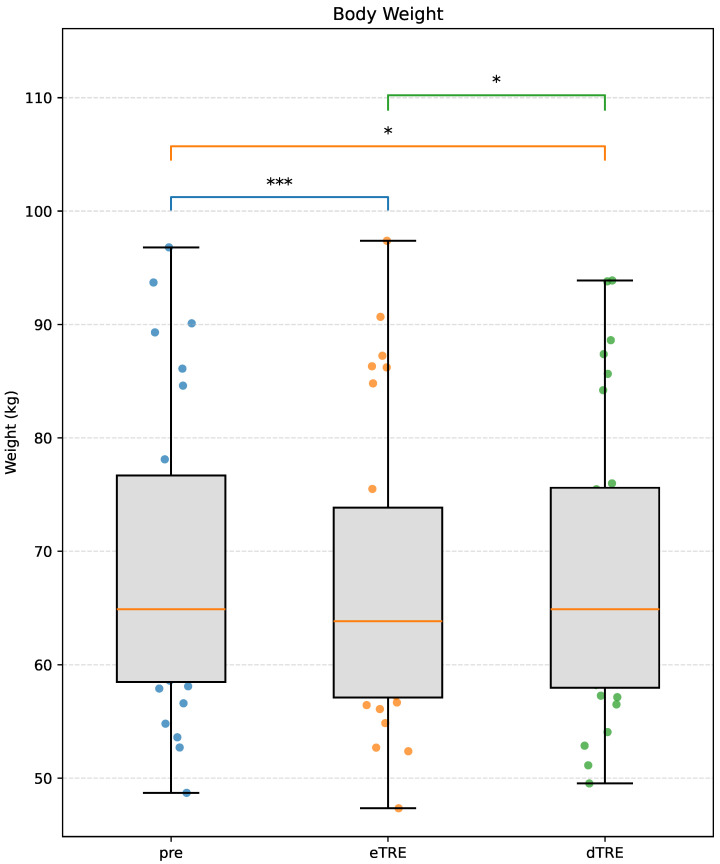
Changes in body weight across the three phases: Pre, eTRE, and dTRE. Data are presented as box plots with individual data points overlaid. The orange line within each box represents the median. Asterisks indicate statistical significance (* p<0.05, *** p<0.001) based on linear mixed-effects models with Bonferroni-adjusted post hoc comparisons.

**Figure 3 nutrients-17-03011-f003:**
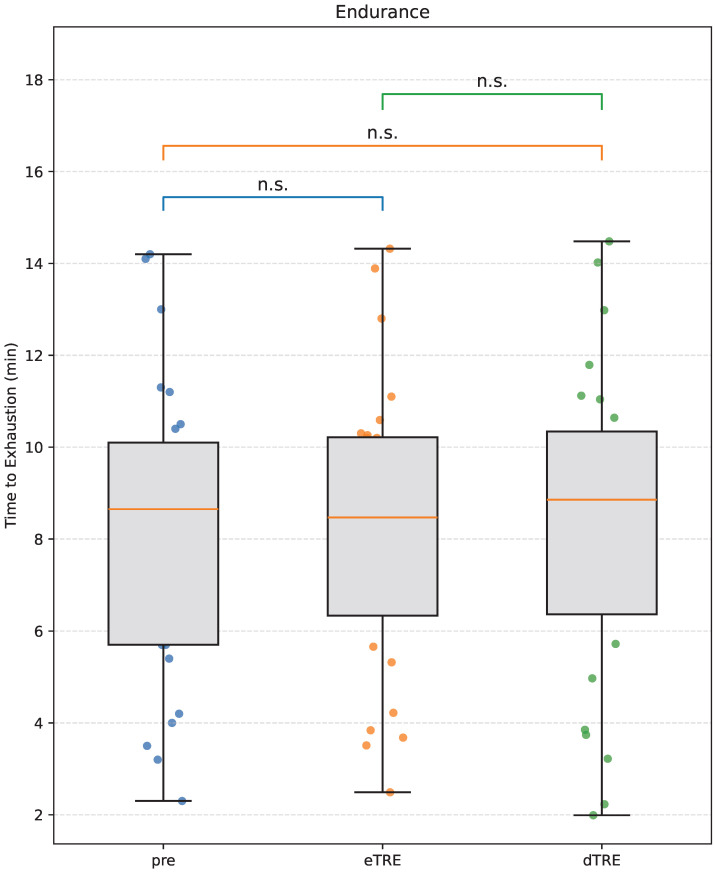
Changes in aerobic endurance across the three phases: Pre, eTRE, and dTRE. Data are presented as box plots with individual data points overlaid. The orange line within each box represents the median. Asterisks indicate statistical significance (n.s., not significant) based on linear mixed-effects models with Bonferroni-adjusted post hoc comparisons.

**Figure 4 nutrients-17-03011-f004:**
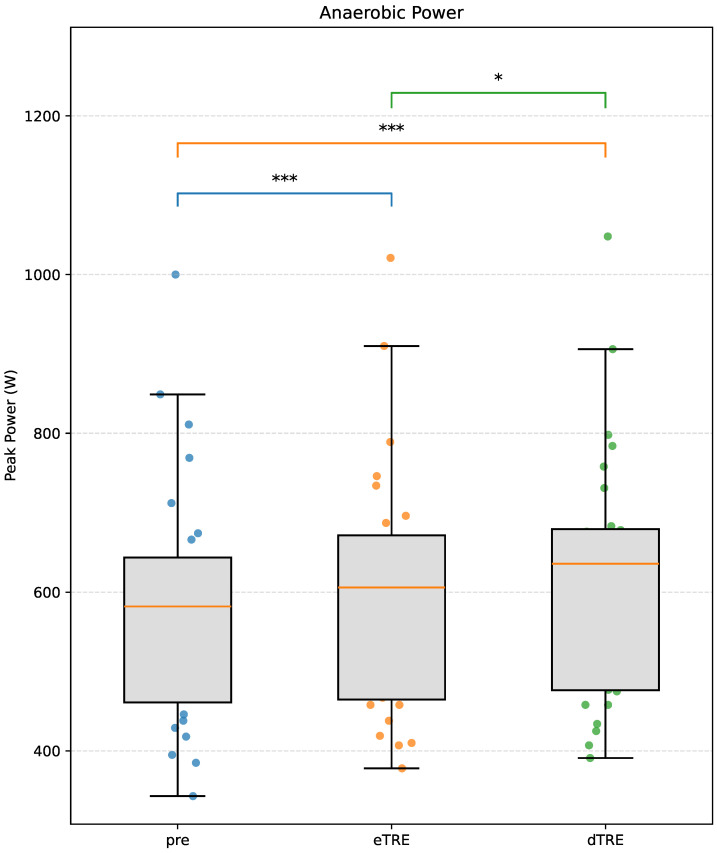
Changes in anaerobic power across the three phases: Pre, eTRE, and dTRE. Data are presented as box plots with individual data points overlaid. The orange line within each box represents the median. Asterisks indicate statistical significance (* p<0.05, *** p<0.001) based on linear mixed-effects models with Bonferroni-adjusted post hoc comparisons.

**Table 1 nutrients-17-03011-t001:** Participant characteristics at baseline (N = 28; 16 women).

Variable	Mean ± SD
Age (years)	23.47 ± 2.87
Body Weight (kg)	68.71 ± 13.52
Height (cm)	171.45 ± 10.16
BMI (kg/m^2^)	23.37 ± 5.37
Aerobic Endurance (min)	8.22 ± 3.21
Anaerobic Power (W)	581.79 ± 152.93

Values are presented as mean ± SD. Aerobic endurance = cycling time to exhaustion at a fixed workload; anaerobic power = peak power output during sprint cycling.

**Table 2 nutrients-17-03011-t002:** Summary of changes in body weight, aerobic endurance, anaerobic power, and daily energy intake across the three phases (mean ± SD, *p*-value, and Cohen’s $d_{z}$).

Variable	Phase	Mean ± SD	Comparison	*p*-Value	Cohen’s $d_{z}$
Body Weight (kg)	Pre	68.71 ± 13.52	Pre vs. eTRE	<0.001 ***	1.17
	eTRE	67.15 ± 13.35	Pre vs. dTRE	0.022 *	0.55
	dTRE	68.10 ± 13.22	eTRE vs. dTRE	0.020 *	0.56
Aerobic Endurance (min)	Pre	8.22 ± 3.21	Pre vs. eTRE	n.s.	—
	eTRE	8.23 ± 3.13	Pre vs. dTRE	n.s.	—
	dTRE	8.33 ± 3.36	eTRE vs. dTRE	n.s.	—
Anaerobic Power (W)	Pre	581.79 ± 152.93	Pre vs. eTRE	<0.001 ***	1.10
	eTRE	603.04 ± 153.41	Pre vs. dTRE	<0.001 ***	1.20
	dTRE	617.21 ± 156.83	eTRE vs. dTRE	0.025 *	0.54
Energy Intake (kcal)	Pre	2163.96 ± 513.66	Pre vs. eTRE	<0.001 ***	0.51
	eTRE	1902.57 ± 525.62	Pre vs. dTRE	<0.001 ***	0.46
	dTRE	1967.04 ± 567.11	eTRE vs. dTRE	0.040 *	0.12

Values are presented as mean ± standard deviation (SD). *p*-values are derived from Bonferroni-adjusted paired-sample *t*-tests following a significant main effect of phase (tested via linear mixed-effects models). Cohen’s $d_{z}$ is reported for statistically significant comparisons only. n.s., not statistically significant (*p* > 0.05); * *p* < 0.05; *** *p* < 0.001.

## Data Availability

The dataset supporting this study is openly available in Zenodo at https://doi.org/10.5281/zenodo.16874956.

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
