# Peer review of "Time-Restricted Eating Without Exercise Enhances Anaerobic Power and Reduces Body Weight: A Randomized Crossover Trial in Untrained Adults"

_nutrients, 2025, doi:10.3390/nu17183011_

Round 1

Reviewer 1 Report

Comments and Suggestions for Authors

Please see attached PDF with comments

Reviewer 2 Report

Comments and Suggestions for Authors

Comments to authors

This study examined the impact of time-restricted feeding on various outcomes in untrained adults, including body weight, aerobic endurance and anaerobic performance, without the addition of exercise. While the study is interesting, some improvements and changes are needed.

Abstract

  • In addition to the total number of participants, authors should include the number of women in the abstract. They should also include the mean or median age and standard deviation (SD) or interquartile range (IQR), depending on the distribution of the data.

  • “Results: Body weight significantly 20 decreased after eTRE (−1.56 kg, 95% CI [−2.07, −1.05], p < 0.001, Cohen’s d_z = 1.17) and 21 dTRE (−0.61 kg, 95% CI [−1.12, −0.10], p = 0.022, Cohen’s d_z = 0.55), with a greater reduc- 22 tion observed during eTRE.”: The authors should include, in addition to the pre-post differences for each intervention group, the difference in differences (I don't know if I'm making myself clear). That is, if in group 1 the effect was -1.56 kg (with a specific 95% CI) and in group 2 the effect was -0.61 kg (with another 95% CI), what was the difference between those groups? With their 95% CI.

Introduction

  • “Therefore, this study aims to fill this knowledge gap by systematically evaluating 118 whether time-restricted eating (TRE)”: TRE has been defined previously. Check for abbreviations that have been defined in multiple places.

  • In the introduction, summarise the effect of TRE on trained subjects. Also summarise the effect of Ramadan on trained subjects. This will lengthen the introduction and undermine the focus on untrained subjects. If it is necessary to include this information, it should be done in a much more concise form. In fact, I would argue that it would be more effective to summarise this information in the introduction and then explain it in more detail in the discussion.

  • I seem to recall that references in Nutrients are numbered in brackets.

Methods

  • In general, studies adhere to recommendations or guidelines such as PRISMA, CONSORT, STROBE, etc. The guidelines that your study followed should be included at the beginning of the 'Methods' section. A checklist of these guidelines should also be included in the supplementary material.

  • It is important to provide a brief explanation of how the randomisation was performed, as Cochrane's RoB2 requires this information to avoid potential bias.

  • Figure 1. Could the authors provide a colour figure? This would make the figure more attractive and facilitate its interpretation.

  • I believe the statistical analyses were performed on the entire sample. Did the authors consider presenting the results separately for each sex? I understand that the sample size is limited, but if possible, this would provide more information since what occurs in one sex does not necessarily occur in the other.

  • Did the authors perform a sample size calculation regarding the sample size? Or was the number of participants based on availability? It is important to perform a sample size calculation prior to any analysis to ensure sufficient statistical power.

  • Did the variables follow a normal distribution? At the very least, it is important to specify the distribution of the variables based on the Kolmogorov test (or the Shapiro test in your case). This improves the interpretation of the variables, and sometimes statistical tests are chosen based on their distribution.

Results

  • Table 1: The table would be improved by arranging the variables vertically rather than horizontally. It would also be interesting to include additional variables, such as aerobic endurance and anaerobic power.

  • “238 suggesting that the weight-reducing effect was more pronounced during the eTRE phase. 239 These findings suggest that although both early and delayed TRE protocols led to reduc-240 tions in body weight, the early TRE protocol was more effective in promoting weight loss 241 (Figure 2). 242”: This is an interpretation, therefore, it is not appropriate to be in results, but in discussion.

Discussion

  • As I mentioned earlier, there is a lack of analysis and discussion of the possible effect of this intervention by sex.

  • The results could also be influenced by BMI or percentage of fat (or lean tissue, depending on the perspective). This is something the authors should discuss in greater depth.

  • While I understand that this cannot be changed, one of my biggest concerns is the lack of a control group. Thus, the results obtained refer specifically to the comparison of the two intervention groups (i.e. which is more effective). I also understand that, in the absence of a control group, we can assume that if there is a pre-post difference, there may be an effect. Nevertheless, the authors should discuss how the lack of a control group could have affected the results and their interpretation.

  • Additionally, a crossover design for this type of intervention could present a problem. If participants receive both interventions, it is possible that they will acquire metabolic changes upon receiving the first intervention that persist until the second intervention. The authors have pointed this out in the limitations section: 'Second, although a washout period was implemented, residual effects between phases cannot be ruled out and no new baseline was established.' However, it would be helpful if they could explain how this could influence the results in more detail.

Reviewer 3 Report

Comments and Suggestions for Authors

Dear,

Thank you for your work. Im going to do some advices for improve the paper. 

The citations are in the wrong format according to the journal guidelines; they should be numbered in brackets in order.
Once an acronym has been defined, it is not necessary to redefine it later. For example, in line 119, “time-restricted eating (TRE)” should not be redefined. Review throughout the document.
Was the sample randomized for the formation of each group? How was this done?
How were consistent lifestyle habits (sleep schedules, physical activity levels, and diet composition) controlled and monitored? This should be specified in detail.
Regarding the measures used, it is necessary to cite the authors who designed or used these anaerobic power and aerobic endurance tests.
In the discussion, the limitations noted should be taken into account when discussing the findings.
A section in the discussion referring to the practical implications suggested by the research findings is necessary.

Round 2

Reviewer 2 Report

Comments and Suggestions for Authors

Comments to authors

The authors were very receptive to the comments and suggestions that were made.